# Precursor Intensity-Based Label-Free Quantification Software Tools for Proteomic and Multi-Omic Analysis within the Galaxy Platform

**DOI:** 10.3390/proteomes8030015

**Published:** 2020-07-08

**Authors:** Subina Mehta, Caleb W. Easterly, Ray Sajulga, Robert J. Millikin, Andrea Argentini, Ignacio Eguinoa, Lennart Martens, Michael R. Shortreed, Lloyd M. Smith, Thomas McGowan, Praveen Kumar, James E. Johnson, Timothy J. Griffin, Pratik D. Jagtap

**Affiliations:** 1Department of Biochemistry, Molecular Biology and Biophysics, University of Minnesota, Minneapolis, MN 55455, USA; caleb_easterly@med.unc.edu (C.W.E.); rsajulga@umn.edu (R.S.); kumar207@umn.edu (P.K.); tgriffin@umn.edu (T.J.G.); 2Department of Chemistry, University of Wisconsin, Madison, WI 53706, USA; rmillikin@wisc.edu (R.J.M.); mshort@chem.wisc.edu (M.R.S.); smith@chem.wisc.edu (L.M.S.); 3VIB-UGent Center for Medical Biotechnology, VIB, Ghent University, 9000 Ghent, Belgium; a.argentini@gmail.com (A.A.); ignacio.eguinoa@psb.vib-ugent.be (I.E.); lennart.martens@vib-ugent.be (L.M.); 4Minnesota Supercomputing Institute, University of Minnesota, Minneapolis, MN 55455, USA; jj@umn.edu (J.E.J.); mcgo0092@umn.edu (T.M.)

**Keywords:** proteomics, label-free quantification, galaxy framework, workflows

## Abstract

For mass spectrometry-based peptide and protein quantification, label-free quantification (LFQ) based on precursor mass peak (MS1) intensities is considered reliable due to its dynamic range, reproducibility, and accuracy. LFQ enables peptide-level quantitation, which is useful in proteomics (analyzing peptides carrying post-translational modifications) and multi-omics studies such as metaproteomics (analyzing taxon-specific microbial peptides) and proteogenomics (analyzing non-canonical sequences). Bioinformatics workflows accessible via the Galaxy platform have proven useful for analysis of such complex multi-omic studies. However, workflows within the Galaxy platform have lacked well-tested LFQ tools. In this study, we have evaluated moFF and FlashLFQ, two open-source LFQ tools, and implemented them within the Galaxy platform to offer access and use via established workflows. Through rigorous testing and communication with the tool developers, we have optimized the performance of each tool. Software features evaluated include: (a) match-between-runs (MBR); (b) using multiple file-formats as input for improved quantification; (c) use of containers and/or conda packages; (d) parameters needed for analyzing large datasets; and (e) optimization and validation of software performance. This work establishes a process for software implementation, optimization, and validation, and offers access to two robust software tools for LFQ-based analysis within the Galaxy platform.

## 1. Introduction

Peptide- and protein-level quantification (either labeled or label-free) is routinely used in mass spectrometry (MS)-based shotgun proteomics data analysis workflows to determine the relative abundance of peptides or proteins in a given sample [1], including post-translationally modified peptides [2] and amino acid sequence variants identified by proteogenomics [3,4]. In the field of metaproteomics, where protein samples obtained from environmental microbiomes are studied, the quantification of microbial peptides or “metapeptides” (peptides obtained from shotgun sequencing of microbial communities) is essential to perform taxonomic and functional quantification of proteins expressed from the microbiome [5]. 

In the case of the label-free quantification (LFQ) methods, the peak intensity or area under the curve of a detected peptide ion allows the relative quantification of peptides across different samples. LFQ [6,7] is a useful method for quantification when the introduction of stable isotopes is impractical (for example, in human or animal model studies) or for applications such as proteogenomics or metaproteomics, which rely on peptide-level quantification. Currently, there are several software packages available for LFQ analysis [8]. LFQ analysis can be performed by public domain software suites such as MaxQuant [9] and Skyline [10], or by commercial software such as PEAKS [11] and Progenesis [12]. Although commercial and actively-supported software offers reliability and ease of use, its usage comes with a cost and usually includes canned features that are used for most standard datasets. Open-source software, on the other hand, has the benefit of being amenable to testing and optimization for emerging disciplines to offer economical options for data analysis. 

In this study, through a rigorous testing and evaluation process, we have incorporated and optimized two established, open-source tools, moFF [13] and FlashLFQ [14] in the Galaxy platform. In order to achieve this, we worked with the software developers of these tools and tested features using two benchmark datasets, the ABRF Proteomics Research Group (PRG) 2015 dataset [15] and a Universal Proteomics Standard (UPS) dataset [1], and compared the outputs with results from MaxQuant, a highly used standalone software platform capable of LFQ analysis. Based on feedback, the tool developers of moFF and FlashLFQ made changes to the software’s capabilities, which included (a) using match-between-runs (MBR); (b) ability to process and analyze large input datasets; (c) compatibility with a variety of input file formats. After this rigorous evaluation and optimization, these tools were implemented in the accessible and reproducible Galaxy [16] platform. Galaxy tools are maintained and developed by an international community of developers (https://galaxyproject.org/iuc/) so as to facilitate ease of usage and maintain its contemporary status for any emerging software tools or applications. An additional advantage of having these tools available via the Galaxy platform is the ability to process the data in workflows, wherein multiple tools can be used in a sequential manner to generate processed outputs from the input data. The Galaxy for proteomics (Galaxy-P) team has developed workflows related to MS-based multi-omic studies such as, proteogenomics [17,18] and metaproteomics [19,20]. The addition of these precursor intensity-based LFQ tools to the existing workflows will facilitate peptide level quantification for multi-omics research studies, as well as more standard proteomics applications. 

As a result of this study, we made two quantitative software tools available to researchers via the Galaxy platform. These software are available via the Galaxy Tool Shed [21,22], GitHub, and on Galaxy public instances.

## 2. Methods

We used two datasets, (a) an ABRF dataset [15] and (b) a spiked-in benchmark UPS dataset [1] to determine the accuracy of each tool with regards to their calculated protein fold-changes. We obtained the MS (raw) data from publicly available repositories and converted them to MGF (Mascot generic format) files using MSConvert (vendor support) (Galaxy Version 3.0.19052.0) [23] to make it compatible with search algorithms within the Galaxy Platform. moFF and FlashLFQ processing were performed within the Galaxy platform (Version 19.09).

### 2.1. (A) ABRF Dataset

The spiked-in dataset from the ABRF PRG 2015 study was used to determine the accuracy of each software tool. This dataset, generated through the collaborative work of the ABRF Proteomics Research Group (https://abrf.org/research-group/proteomics-research-group-prg) contains four proteins added to human cell lysate samples: ABRF-1 (beta galactosidase from *Escherichia coli*), ABRF-2 (lysozyme from *Gallus gallus*), ABRF-3 (amylase from *Aspergillus niger*) and ABRF-4 (protein G from *Streptococcus*) [15]. Each sample contained the four proteins at the same concentration, while the concentrations varied across the four samples: 0 (blank/negative control), 20, 100, and 500 fmol. The peptide raw data for the ABRF dataset was acquired on the LTQ orbitrap Velos with EASY-nLC.

### 2.2. (B) Spiked-In UPS Benchmark Dataset

To evaluate these tools, we downloaded publicly available data [1] (PRIDE #5412; ProteomeXchange repository PXD000279), wherein UPS1 and UPS2 standards (Sigma-Aldrich, St. Louis, MO, USA) were spiked into *E. coli* K12 strain samples. Based on the dynamic benchmark dataset protocol, the UPS and *E. coli* peptides we quantified using nanodrop spectrophotometer at 280 nm, and 2 μg of *E. coli* peptides were spiked with 0.15 μg of UPS1 or UPS2 peptides. About 1.6 μg of the mix was analyzed on the Q Exactive (Thermo Fisher, Waltham, MA, USA) mass spectrometer [1]. The UPS1 and UPS2 standards contain 48 human proteins at either the same (5000 fmol, UPS1) or varying concentration (50,000 fmol to 0.5 fmol, UPS2), respectively.

### 2.3. Peptide Identification

For both datasets, we used SearchGUI (SG) [24] (version 3.3.3.0) and Peptide Shaker (PS) [25] (version 1.16.26) to search the MS/MS spectra against respective protein FASTA databases along with contaminants from cRAP database (https://www.thegpm.org/crap/). Although SearchGUI has the option to use as many as eight search algorithms, we used only four search algorithms (X!tandem, OMSSA, MSGF+, and Comet) for this evaluation study. 

For the spiked-in ABRF PRG dataset, a protein FASTA file was generated by merging the UniProt human reference database with spiked-in proteins and contaminant proteins (73,737 protein sequences database generated on 6 February 2019). Search parameters used were trypsin enzyme for digestion, where two missed cleavages were allowed. Carbamidomethylation of cysteine was selected as a fixed modification and methionine oxidation was selected as a variable modification. The precursor mass tolerance was set to 10 ppm and the fragment mass tolerance to 0.5 Da, with minimum charge as 2 and maximum charge of 6. For Peptide Shaker, the false discovery rate (FDR) was set at 1% at the PSM, peptide, and protein level, along with filtering the peptide length ranging from 6–65 peptides.

For the spiked-in UPS dataset, the mass spectra were searched against a protein FASTA database provided by Cox. et al., 2014 [1], (4494 protein sequences database generated on 25 July 2019). The parameters for SearchGUI-Peptide Shaker analysis were as follows: precursor mass tolerance was set to 10 ppm and the fragment mass tolerance to 20 ppm with minimum and maximum charge as 2 and 6, respectively. 

For MaxQuant analysis (version 1.6.7.0, Cox lab, Max Planck institute of Biochemistry, Martinsried, Germany), the built-in Andromeda search engine [26] was used. The parameters for MaxQuant were matched with the SearchGUI-PeptideShaker search. The fixed modification was set for carbamidomethylation of cysteine and oxidation of methionine as a variable modification. The FDR was set at 1% and the MS/MS tolerance was set at 10 ppm. The tabular output data from Peptide Shaker (PSM.tab) and Andromeda (msms.txt) were used for protein quantification.

### 2.4. Quantification Tools

moFF and FlashLFQ, were initially tested outside of the Galaxy platform. We tested various releases for moFF (versions 1.2.0 to 2.0.2) and FlashLFQ (versions 0.1.99 to 1.0.3) and provided developers with feedback to improve software stability and data quality. We then implemented these updated tools within Galaxy. The results from moFF (version 2.0.2) and FlashLFQ (version 1.0.3) were then compared with MaxQuant (version 1.6.0.16), a widely used LFQ quantification software suite. For testing, all the quantification tools were set at monoisotopic tolerance of 10 ppm and run with or without MBR, where indicated. 

### 2.5. Normalization and Protein Quantification

After peptide-level precursor intensity values were generated, normalization was performed using limma [27], and peptides were summarized into protein-level abundances with protein expression control analysis (PECA) [28]. Specifically, the “normalizeBetweenArrays” limma function was used for most normalization methods (i.e., scale, cyclic loess, and quantile). For VSN (variance stabilizing normalization), the “normalizeVSN” limma function was used [27,29]. After normalization, PECA was used to combine the peptide-level measurements to protein-level values for the detection of differentially expressed proteins. These two tools were run via custom R scripts (version 1.3), which can be accessed via the Appendix A (https://github.com/galaxyproteomics/quant-tools-analysis).

## 3. Results

Both moFF and FlashLFQ are established software tools and contain useful features such as amenability to Galaxy implementation, compatibility with existing Galaxy upstream and downstream tools, ability to read mzML and Thermo raw file formats, open-source code, MBR functionality, and results that can be easily evaluated with performance metrics. 

moFF is an extensible quantification tool amenable to any operating system. The input for moFF is peptide search engine output and Thermo raw files and/or mzML files; it performs both MS/MS as well as MBR quantification. moFF tool also has a novel filtering option for MBR peak intensities [30]. moFF has been wrapped in Galaxy (Figure 1A) using a Bioconda package [31]. The Galaxy version of moFF is available via Galaxy toolshed [21], GitHub [32] and Galaxy public instances (proteomics.usegalaxy.eu, usegalaxy.be and z.umn.edu/metaproteomicsgateway). 

FlashLFQ is a peptide and protein LFQ algorithm developed for proteomics data analysis. It was developed to quantify peptides and proteins from any search tool, including MetaMorpheus, which also performs PTM identification from MS/MS data. It uses Bayesian statistics to estimate the difference in the abundance of inferred proteins between samples, though this feature was not evaluated here. FlashLFQ can normalize fractionated datasets by using a bounded Nelder–Mead optimizer [33] to find a normalization coefficient for each fraction, similar to MaxLFQ. FlashLFQ was implemented in Galaxy (Figure 1B) within a Singularity container [34] as FlashLFQ is a Windows application requiring the NET core framework for deployment in the Unix-based Galaxy environment. Singularity provides a secure means of running such tools in Galaxy. The Galaxy version of FlashLFQ is available via Galaxy toolshed [22] GitHub [35] and via Galaxy public instances (proteomics.usegalaxy.eu and z.umn.edu/metaproteomicsgateway).

Relevant features of moFF and FlashLFQ, as well as the design of the evaluation study, are summarized in Figure 2. An essential aspect of this study was both of these tools being in active development by groups amenable to collaboration, which greatly helped optimization-based tests and feedback from the Galaxy-P team members.

To generate peptide identification inputs for moFF and FlashLFQ, datasets were searched against appropriate protein databases using SearchGUI/PeptideShaker. The speed and accuracy of FlashLFQ and moFF were evaluated in comparison to MaxQuant, a popular software tool used for LFQ. For MaxQuant, searches were performed by MaxQuant’s built-in Andromeda search algorithm. All three software programs have an MBR feature, where unidentified peaks are “matched” to identify peaks in other runs based on similar *m/z* and retention time. MaxLFQ, an algorithm within MaxQuant, normalizes raw intensities, and also aggregates them into protein groups [1]. For moFF and FlashLFQ, limma was used to normalize peptide intensities, and PECA was used to determine protein fold-changes and associated *p*-values. The limma tool within Galaxy implements different normalization techniques such as quantile, VSN, cyclic LOESS, and scale normalization. Users can choose between these normalization methods. FlashLFQ also has built-in normalization and protein quantification functions, which we have used in this study. 

After ascertaining that moFF and FlashLFQ results correlate well with MaxQuant results (Appendix A), we set out to evaluate the MBR feature of the software tools. For this, we used the ABRF PRG datasets, with four spiked-in proteins at three different concentrations (20 fmol, 100 fmol, and 500 fmol) and a negative control (see methods). The spiked-in proteins should not be detected in the negative control, either with or without MBR. We observed that moFF and FlashLFQ outputs showed non-zero intensity values for the spiked-in proteins in blank control samples if MBR was enabled (Figure 3A, left). The match-between runs (MBR) module for the earlier version of both the software, moFF and FlashLFQ, simply predicted the retention time for the matched peptide in the target run by looking for intensity in the target *m/z* and RT window. If the target peptide was not present in that run, overlapping eluting peptides or noise would result in the assignment of spurious background signals. As a result, ABRF spiked in proteins were detected in the blank control sample whole using the MBR mode in the earlier versions of the software tools. We worked with the developers to improve their MBR algorithms so that intensity values for these proteins in the blank control samples were correctly reported as zero (Figure 3B, right). moFF’s new version removes spurious matches by filtering scans for the similarity between the theoretical isotopic envelope of target peptide and corresponding envelope found in the target run. This method has been published in the moFF 2.0 version [30]. The FlashLFQ developers implemented an optional setting that requires a protein to have at least one peptide assigned to an MS/MS spectrum in a sample group so that a non-zero value can be assigned to the peptides in the sample group. Due to the above-mentioned changes in the algorithms and subsequent filtering steps, the newer versions of these tools do not detect spurious signals for target peptides in blank control samples.

The 500 fmol and 100 fmol datasets from the ABRF dataset were used to determine the fold-change accuracy (Figure 3B). In order to determine the accuracy of the fold-change, root mean squared log error (RMSLE) [36] was calculated,
(1)$$RMSLE=\sqrt{\frac{\sum_{i=1}^{N}{\left(log_{10}\;r_{i}-log_{10}\hat{r_{i}}\right)}^{2}}{N}\;}$$
where, *r_i_* is the true ratio,$\;{\hat{r}}_{i}$ is the estimated ratio and *N* is the number of proteins identified via sequence database searching of the sample.

Root mean squared log error is a metric to evaluate the difference between predicted and observed values. In this case, the values being compared are the predicted (known) fold-changes and the observed fold-changes. RMSLE being an error based metric provides the true picture of prediction quality, however, deciding a suitable threshold value is challenging. The objective was to obtain an RMSLE value closer to zero for all the tools. The RMSLE values for the three tools are shown in Figure 3B, with the MBR feature enabled. We observed that moFF with MBR had slightly higher error compared to the other tools. MaxQuant’s MBR and FlashLFQ’s MBR perform quite similarly, though all three tools show a low error when the MBR is enabled. 

Although MaxLFQ and FlashLFQ have their own in-built methods for normalizing peptide abundance, for a more direct comparison of moFF and FlashLFQ performance, we normalized the peptide intensity levels using limma and obtained differentially expressed proteins through the PECA bioconductor package. Normalized peptide intensity values from moFF and FlashLFQ were input into PECA, wherein, the tool calculates the *p*-values of the peptide level data and then groups the values into protein level data. The PECA output was then compared with MaxQuant values using the UPS benchmark dataset. For this, quantitative information for the 48 proteins from the UPS dataset was extracted using an R-script to generate a tabular output with fold-change values. 

The fold-change accuracy of all quantified UPS proteins after normalization was calculated by comparing the estimated protein fold-change with the true fold-change using the RMSLE (Figure 4A). 

Figure 4A shows the comparison of different normalizations using the MBR values. Although the bar graph shows that MaxQuant’s MaxLFQ performed the best compared to all, it did so at the cost of the number of proteins quantified. However, we noticed that MBR from FlashLFQ, with its in-built normalization (light blue bar in Figure 4A), performed better overall in terms of quantification and the number of proteins identified. Meanwhile, moFF and FlashLFQ provided higher numbers of quantified proteins while still maintaining low RMSLE values. We also performed comparison studies of MBR vs. no MBR, the results of which are shown in Appendix A. We also found that moFF and FlashLFQ quantified similar numbers of peptides across the ABRF and UPS datasets (Appendix A).

After evaluating the RMSLE for all proteins, we estimated the accuracy of similarly abundant UPS proteins (Figure 4B). We categorized the UPS standards by their concentration ratio (UPS2/UPS1), which resulted in 6 different categories (i.e., ratios of 1 to 0.0001). The results showed that MaxQuant quantification works optimally for high and medium abundant proteins. However, for low abundance proteins, the fold-accuracy was lower, presumably because of missing intensity values. Another important observation was that MaxLFQ denotes a smaller error compared to the other tools in the low abundance samples, but quantifies fewer proteins. An evaluation study for MBR vs. noMBR was also performed and showed a similar trend represented in Appendix A. 

After evaluation of the moFF and FlashLFQ tools, we worked with the developers of these tools to implement their optimized software in Galaxy, enabling integration into diverse MS-based proteomics workflows and promoting their usage by the Galaxy community. Our implementation will allow the users to choose their choice of tool and normalization (Appendix A), which will benefit their research. 

## 4. Discussion

Protein and peptide-level quantification has been used by proteomics researchers to determine how the proteome responds to biological perturbation [37]. In particular, precursor-intensity based LFQ has enabled researchers to perform quantitative proteogenomics analyses [38]. Quantitative changes in the proteome can also be correlated with transcript abundance changes [39] to get a more complete picture of how an organism responds to a stimulus. For example, in cancer proteogenomics studies, these abundance measurements help identify differential expression patterns of variant peptides that may have functional significance in cancer [40,41].

Peptide-level quantification also aids in functional studies of microbial communities and microbiomes using metaproteomics. For example, in metaproteomics studies, metapeptides or metaproteins detected from environmental [42] or host-derived samples [43] can be quantified to shed light on the dynamics of the taxa, biological function, and their abundance [44]. Our group has developed and optimized Galaxy-based tools and workflows for proteogenomics [45] and metaproteomics analyses [19,20]. Tools implemented through this study will extend these workflows to enable quantification of metapeptides and/or metaproteins.

In our analyses, the three LFQ tools, moFF, FlashLFQ, and MaxQuant, correlate well in their results according to our evaluation. In this study, we have added moFF and FlashLFQ to the Galaxy framework, which not only facilitates the dissemination of these tools but also enables automated data analysis by using them within workflows [16]. We also highlight the importance of the process of careful user evaluation, feedback to developers, and optimization of the tools and workflows. Preliminary testing was performed on the command line or GUI versions of the tool. These tools were then packaged into the Galaxy platform, where results were compared to the command-line/GUI versions and also optimized more usage in automated workflows. 

Open-source software usage has faced challenges due to dependencies such as operating system (Windows, Linux, OSX), language or platforms (Python, C++, Java), lack of adhering to HUPO standards [46], and installation or usability issues [47]. To overcome these issues, the Galaxy-P project, as well as others in the Galaxy community, have sought collaborations with many research groups that have developed these tools, following a protocol which includes defining key input and output data types, establishing key operating parameters for the Galaxy tool, overcoming operating system compatibility issues (e.g., Singularity containers for Windows tools), along with rigorous testing and optimization. This collaborative and iterative process of development and optimization ensures the software performs accurately and efficiently within the Galaxy platform.

Ideally, software tools that are UNIX based, such as moFF, are easier for deployment within Galaxy. We also demonstrated here that tools such as FlashLFQ could be packaged within a singularity container to enable easy and secure implementation within Galaxy. MaxQuant, which is a popular, public-domain proteomics software package, is available in both Windows and Linux-compatible versions [9]. Although in the early development and testing phase, the LFQ module that uses MS1 precursor intensity data within MaxQuant (MaxLFQ) was made available within Galaxy toolshed [48]. Once fully tested and evaluated, accessibility to this software via the Galaxy platform will offer even more choices for precursor-intensity based quantification. Offering users a choice of multiple validated software tools also highlights the benefits of a workflow engine such as Galaxy, where users can easily develop parallel workflows using different combinations of tools to determine methods that provide optimal results based on user requirements.

In our tests, we observed that FlashLFQ has a faster runtime as compared to the other two tools. MaxQuant processing time is longer, presumably since it performs peptide identification and quantitation simultaneously. For example, on the same computing device, the UPS dataset was processed by FlashLFQ in approximately 15 min for quantification only, whereas MaxQuant and moFF took 34 min and 3 h, respectively. Our evaluation and availability of these tools within a unified platform such as Galaxy offers users a choice for their workflows where the speed of analysis can also be considered.

## 5. Conclusions 

This study demonstrates a successful collaborative effort in software tool development and dissemination, which is a hallmark of the Galaxy community and the Galaxy-P project [19]. This community-driven approach brings together users and software developers who work together to validate and make the tool accessible and usable for other researchers across the world. The study described here provides a model of success for the process used to ultimately provide optimized, well-validated tools for community use. We did not seek a goal to determine the single best tool for LFQ use, but rather focused on offering users a choice of validated quantification tools amenable to customizable analytical workflows. In addition to our work here, others from the Galaxy community are also working on integrating tools within the MaxQuant suite [49], which will extend the choices for LFQ quantification available. As a result of this study, Galaxy users can now confidently use two rigorously validated LFQ software tools (moFF and FlashLFQ) for their quantitative proteomic studies. We are currently working on incorporating the quantitative capabilities of moFF and FlashLFQ within existing metaproteomics and proteogenomics workflows, so that they can be used by the research community in their quantitative multi-omics studies.

## Figures and Tables

**Figure 1 proteomes-08-00015-f001:**
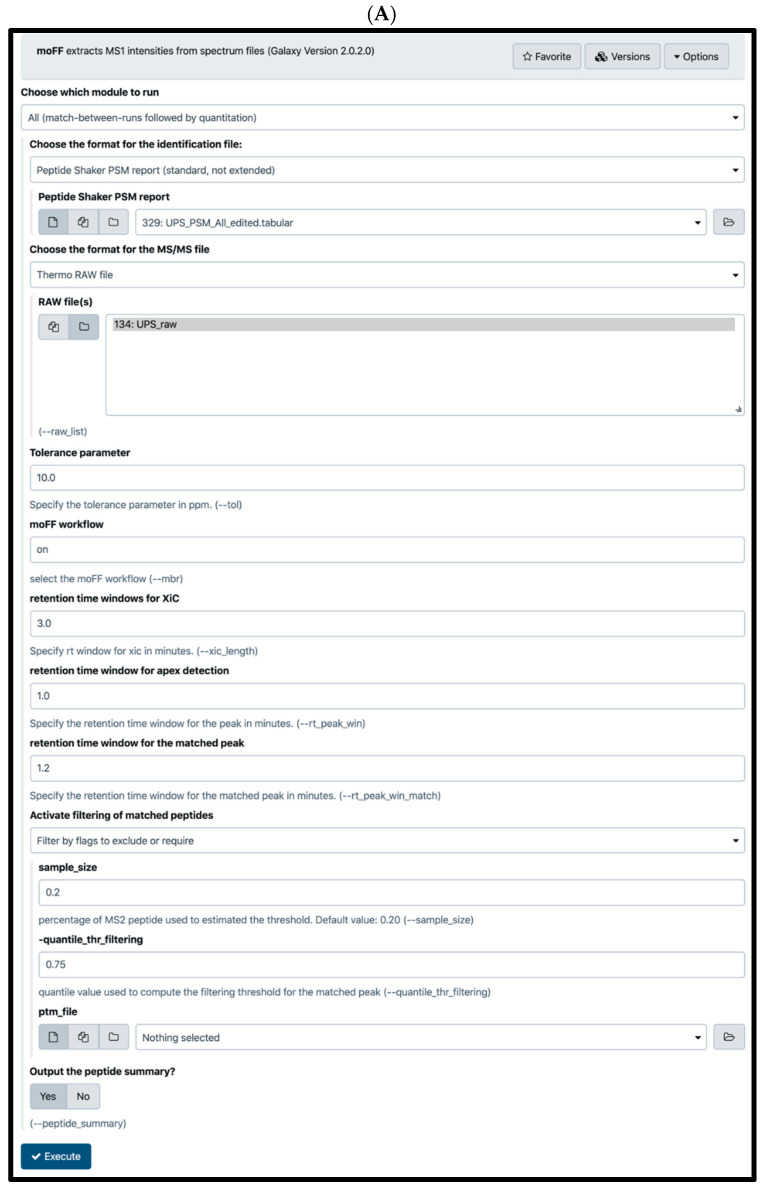
Galaxy interface of moFF and FlashLFQ: (**A**) Bioconductor package of moFF is wrapped within Galaxy and available via Galaxy toolshed [21] and Galaxy public instances (proteomics.usegalaxy.eu). (**B**) A docker/singularity container of FlashLFQ is wrapped within Galaxy and available via Galaxy toolshed [22] and Galaxy public instances (proteomics.usegalaxy.eu).

**Figure 2 proteomes-08-00015-f002:**
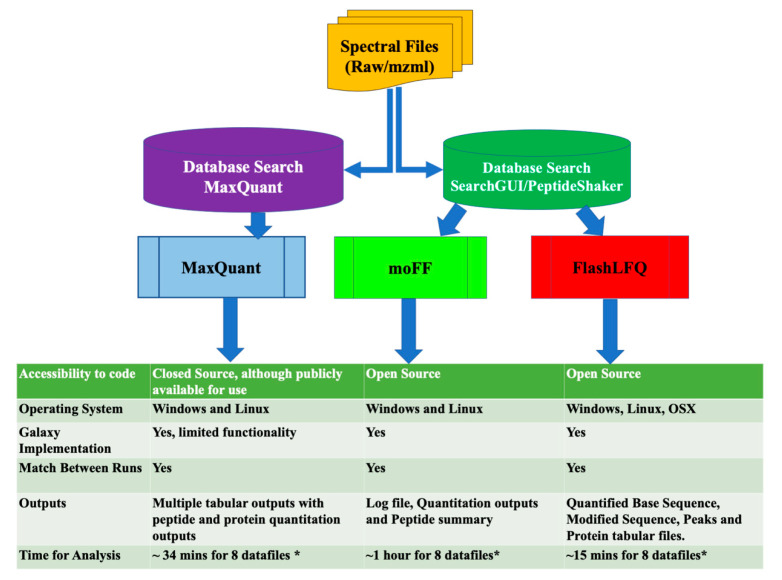
Experimental design of the evaluation study: spectra files are converted to MGF before mass spectra are matched with peptides using respective search engines. Each of the quantification tools use RAW files and the peptide identification tabular output as inputs. The figure also shows the features of each tool. The outputs from all of the tools were then compared against each other. The asterisk symbol (*) denotes that the files were run on same computing device.

**Figure 3 proteomes-08-00015-f003:**
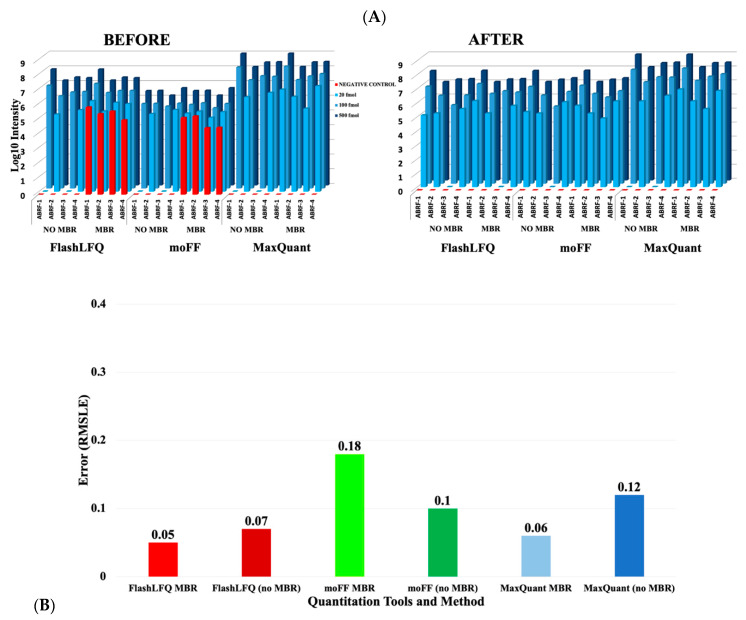
(**A**) Effect of MBR after software version updates: The log10 values of the intensities (blue bars) from each of the four ABRF spiked-in proteins (ABRF-1: beta Galactosidase from *E. coli*, ABRF-2: Lysozyme from *Gallus gallus*, ABRF-3: amylase from *Aspergillus*, ABRF-4: protein G *Streptococcus*) were plotted. The results from prior versions of moFF (v1.2.1) and FlashLFQ (v0.1.99) (before) shows that MBR detects ABRF proteins (shown in red) in the negative control sample in both software. The results from the current versions of moFF (v2.0.2) and FlashLFQ (v1.0.3.0) implemented in Galaxy (after), shows that the MBR feature does not detect ABRF proteins in the negative control. (**B**) Accuracy of fold-change estimation: for evaluating the accuracy of quantified results, we estimated the fold change of the spiked-in proteins in the 500 fmol sample as compared to 100 fmol sample. The root mean squared log error (RMSLE) was calculated for fold change estimation. For this dataset, moFF with MBR displayed significantly higher RMSLE value, whereas FlashLFQ’s MBR performed similarly to MaxQuant’s MBR.

**Figure 4 proteomes-08-00015-f004:**
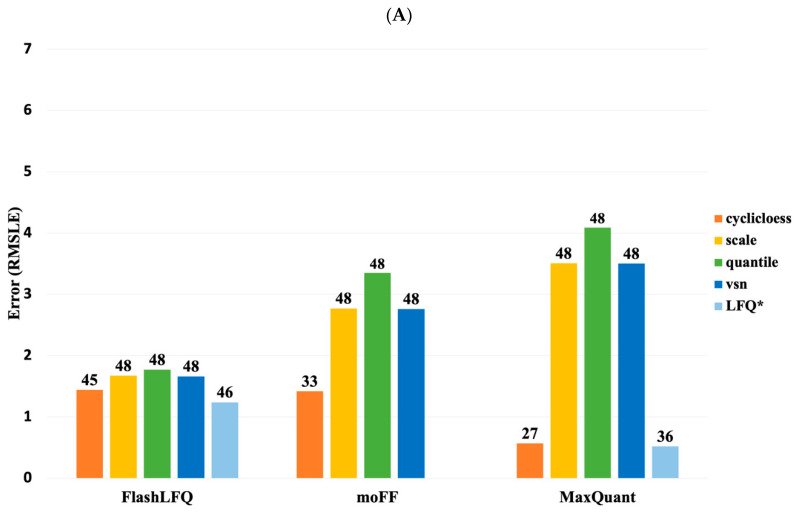
(**A**) Fold-change accuracy (MBR) of all proteins: after normalization, the estimated protein abundance ratios for all the identified UPS proteins were compared to the true abundance ratios, using the root mean squared log error (RMSLE). The plot represents the RMSLE values using different normalization methods. *LFQ denotes the LFQ values represent FlashLFQ’s and MaxQuant’s inbuilt normalization value. The value on the top of the bars denotes the number of proteins that were quantified. (**B**) Fold change accuracy (MBR) of proteins with similar estimated ratios: In total there are 48 UPS proteins, we classified the UPS proteins into different groups based on the UPS2/UPS1 ratio estimation, the true ratios run from 10 to 10^−4^. The value on the top of the bars denotes the number of proteins that were quantified using each normalization method. The RMSLE of the intensity ratio was used to measure the accuracy of the estimated fold changes.

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
