# Peer review of "Precursor Intensity-Based Label-Free Quantification Software Tools for Proteomic and Multi-Omic Analysis within the Galaxy Platform"

_proteomes, 2020, doi:10.3390/proteomes8030015_

Round 1

Reviewer 1 Report

This is an exceptional crafted manuscript and I cannot find a single fault in it. The authors are to congratulated.

I do have a single question. What changes to the software were required to resolve the MBR problem shown in figure 3A? What was causing this issue? If there is one thing that could be added to the manuscript, it is an explanation of what caused this issue.

Author Response

Reviewer Comment: This is an exceptional crafted manuscript and I cannot find a single fault in it. The authors are to congratulated.

I do have a single question. What changes to the software were required to resolve the MBR problem shown in figure 3A? What was causing this issue? If there is one thing that could be added to the manuscript, it is an explanation of what caused this issue.

Author: We thank the reviewer for the comments. The following explanation for the changes made in the software has been added to the revised manuscript. (page number 8 and line numbers 206-211 and 213-220 and highlighted in Yellow)

“ The match-between-runs (MBR) module for the earlier versions of both the software - moFF and FlashLFQ - simply predicted the retention time for the matched peptide in the target run by looking for an intensity in the target m/z and RT window. If the target peptide was not present in that run, overlapping eluting peptides or noise would result in assignment of spurious background signals. As a result, ABRF spiked in proteins were detected in the blank control sample while using the MBR mode in the earlier versions of the software tools.

moFF’s new version removes spurious matches, by filtering scans for similarity between the theoretical isotopic envelope of a target peptide and the corresponding envelope found in the target run.This method has been published in the moFF 2.0 version (J. Proteome Res. 2019, 18, 2, 728–731;Publication Date:December 4, 2018).

FlashLFQ developers implemented an optional setting that requires a protein to have at least one peptide assigned to a MS/MS spectrum in a sample group, so that a non-zero value can be assigned to the peptide in the sample group.

Due to the above changes in the algorithms and subsequent filtering steps, the newer versions of these tools do not detect spurious signals for target peptides in blank control samples.”

Reviewer 2 Report

The manuscript entitled “Precursor intensity-based label-free quantification software tools for proteomic and multi-omic analysis within the Galaxy Platform” evaluated moFF and FlashLFQ - two open-source LFQ tools - and implemented them within the Galaxy platform to offer access and use via established workflows.

  1. Authors should mention the version of each software used in the study. Ex. MSConvert
  2. What is the reason to select fragment mass tolerance to 0.5 Da for ABRF PRG dataset whereas fragment mass tolerance to 20 ppm for Spiked-in UPS dataset?
  3. Authors should provide additional information about the datasets used in the study (ex. Which instrument used etc)
  4. Authors should have used other open source platforms (Skyline, trans proteomics pipeline etc) to compare with moFF and FlashLFQ just like the authors did with MaxQuant

Author Response

1. Authors should mention the version of each software used in the study. Ex. MSConvert

Thank you for the suggestion, We have added the software version to the manuscript (page number 2 and line numbers 77 and 79 and highlighted in Yellow)

2. What is the reason to select fragment mass tolerance to 0.5 Da for ABRF PRG dataset whereas fragment mass tolerance to 20 ppm for Spiked-in UPS dataset?

The mass tolerances  used for mass spectral data searches are determined by the instrument used to acquire MS/MS data. Selecting optimal tolerance values results in better identification statistics - with respect to spectra, peptides and proteins detected in the sample. The  tolerance values described in our study were used since the ABRF dataset was acquired on a lower resolution mass spectrometer compared to the UPS dataset.

3. Authors should provide additional information about the datasets used in the study (ex. Which instrument used etc)

We have made the suggested change in the newer version of the manuscript (page number 2 and 3 and line numbers 87,88 and 92-95 and highlighted in Yellow)

4. Authors should have used other open source platforms (Skyline, trans proteomics pipeline etc) to compare with moFF and FlashLFQ just like the authors did with MaxQuant

This is a good suggestion for comparing multiple tools that perform MS1-based LFQ quantitation. However, for this study we have focussed on a) testing two open-source tools developed by our collaborators (moFF and FlashLFQ); b) comparing their features with a standard LFQ quantitation tool (MaxQuant) and c) eventual implementation of the tools in Galaxy platform.